# Adaptive Non-Singular Terminal Sliding Mode Control Method for Electromagnetic Linear Actuator

**DOI:** 10.3390/mi13081294

**Published:** 2022-08-11

**Authors:** Yingtao Lu, Jiayu Lu, Cao Tan, Maowen Tian, Guoming Dong

**Affiliations:** 1School of Transportation and Vehicle Engineering, Shandong University of Technology, Zibo 255000, China; 2Shandong Zhongbaokang Medical Implements Co., Ltd., Zibo 255000, China

**Keywords:** electromagnetic linear actuator, non-singular terminal sliding mode control, exponential reaching law, friction model, disturbance observer

## Abstract

In this study, to further improve the dynamic performance and steady-state accuracy of the electromagnetic linear actuator (EMLA) for the direct-drive system, the adaptive non-singular terminal sliding mode controller (ANTSMC) based on an improved disturbance observer (IDOB) is proposed. Accurate modeling of the EMLA based on the LuGre friction model was obtained by identification. Aiming at the continuous disturbance of the system, a non-singular terminal sliding mode control based on improved exponential reaching law (IERL) is designed, which is combined with the advantages of fuzzy control. Adaptive adjustment of the IERL gain is achieved to ensure finite time convergence of the tracking error while suppressing system jitter. Based on the IDOB to achieve effective observation compensation under noise disturbance and further improve the system robustness. The stability of the control is analyzed based on the Lyapunov function. The results show that the proposed method suppresses the jitter and improves the tracking performance and robustness of the servo system for different desired trajectories effectively.

## 1. Introduction

In the direct-drive system, linear motors have the advantages of compact size, fast response, high reliability, and low mechanical losses compared to conventional rotary motor drives [1,2]. To meet the rapidly developing requirement in new microrobots, high-precision machine tools, and intelligent vehicles, higher-performance micro linear motors are continuously being investigated [3,4,5].

In recent years, with the rapid development of smart materials, high magnetic energy rare earth permanent magnet materials, manufacturing technology, and advanced control theory. Scholars have further improved the performance of micro linear motors by using highly magnetic materials, designing Halbach arrays instead of conventional pole structures, as well as applying superconducting materials to make windings [6,7,8]. The electromagnetic linear actuators, as an important development direction of linear motors, have become a research hotspot for high-performance linear servo control due to the advantages of cogging-free torque, high linearity, simple structure, and convenient control compared to conventional permanent magnet synchronous linear motors [9,10].

Considering the disturbance of various nonlinear internal and external disturbances in the process of precise actuation. Among them, nonlinear friction has a particularly significant impact on linear motors in terms of high-precision tracking and steady-state positioning. Consequently, it cannot be reduced to a definite constant in the system modeling. An improved least squares algorithm was used by Ma to identify the Stribeck friction model based on a compensation algorithm to achieve better compensation for the friction force [11]. Guc proposed to model and identify the friction based on the Stribeck friction model to reduce the trajectory tracking error based on the streamlined motor model [12]. Nevertheless, there is still some enhancement potential considering the absence of dynamic characteristics of friction in such friction models. Lu introduced improved LuGre friction to model the electromagnetic linear actuator (EMLA), considering the static and dynamic characteristics, such as the Stribeck effect and stick-slip motion during the actuation of the actuator [13]. Shao used an improved artificial fish swarm algorithm (IAFSA) for parameter identification of the LuGre friction model, while the effectiveness of the method was verified by steady-state and dynamic experiments [14]. In summary, a combination of theoretical modeling and model identification based on advanced friction models to achieve refined modeling is becoming a novel trend. However, there remains room for further improvement space based on the long training time of the intelligent control model and the poor real-time performance of the algorithm [15,16].

Linear motors are used as the key execution unit for the realization of high response quality in direct-drive systems. Considering that there is no intermediate buffer mechanism, various nonlinear disturbances act on the linear motor directly, which presents a challenge for the realization of its high precision and high response requirements. Consequently, how to improve the response quality of linear motors under complex and dynamic operating conditions needs to be further investigated. PID, sliding mode control (SMC), active disturbance rejection control, adaptive control, and fuzzy control were successfully applied in micro direct-drive systems [17,18,19,20]. SMC has become an important branch in the field of strong robust and high-precision control. With its robustness, it is widely applied in electro-hydraulic actuators, micro-robots, micro-flying machines, vehicles, and high-precision lathes [21,22]. Nevertheless, the jitter caused by the high gain compensation uncertainty of the SMC can seriously affect the steady-state accuracy and increase power consumption. Jiang proposed a novel reaching law to solve the problem of system jitter, but increased the reaching time [23]. Lu designed a reaching law by saturation function sat(s) instead of a symbolic function to improve the control performance [13]. In addition, other researchers combined the exponential reaching law and the single power reaching law to ensure shorter reaching time and smoother reaching process, but there is still a small jitter under the experiment [24]. Xu proposed a high-order sliding mode observer compensated perturbation to ensure the reduction of the switching term gain to further achieve jitter suppression [25]. Du ensured that the tracking error converges to the origin in finite time by terminal sliding mode control to improve reaching and suppress jitter [26]. Nevertheless, considering the problems of singularity and jitter in the traditional terminal sliding mode control, the SMC with higher response quality for complex working conditions needs to be further improved.

Additionally, in cases of different targets with large unpredictable disturbances, the switching term gain parameter of the SMC would be large, further aggravating the jitter. It is difficult to meet the requirements of high response speed and high tracking accuracy optimal control. Therefore, scholars introduced the disturbance observer (DOB) to compensate for disturbances to improve the robustness of the system while suppressing jitter [27,28,29,30,31]. Wang designed an internal loop observation compensation controller based on discrete SMC to achieve the effective compensation of external disturbances on system performance [27]. Mansouri optimized the two-sided sliding mode observer by a particle swarm algorithm to improve the robustness [28]. Wan proposed the SMC based on an extended state observer, which effectively suppresses the internal and external uncertainties and improves the tracking accuracy significantly [29]. Ye proposed a novel control scheme based on a nonsingular predefined-time sliding-mode manifold [30]. It is shown that the proposed control scheme can produce a good attitude performance even when there exist both parametric and nonparametric uncertainties. In summary, in recent years, many scholars used the complementary advantages of different controls to improve the actuator response performance by improving the reaching law and introducing an observer to suppress system jitter. Nevertheless, higher performance controls need to be further investigated for the optimal control requirements of servo systems with different desired goals under highly complex and dynamic operating conditions.

Based on the above, to improve the high response performance of the EMLA for different desired targets, an adaptive non-singular terminal sliding mode control (ANTSMC) based on an improved disturbance observer (IDOB) is proposed in this study. Firstly, accurate modeling of the EMLA is achieved based on the LuGre friction model obtained from the identification. Then, the improved non-singular terminal sliding mode control (INTSMC) is combined with the advantages of fuzzy control to design the ANTSMC. Furthermore, an IDOB is designed based on the tracking differentiator (TD) to improve the effective compensation performance of the controller under nonlinear disturbances such as noise. The switching gain of the controller is reduced to further suppress the system jitter. Finally, the improvement of the response performance in highly complex operating conditions under different desired targets is achieved through the simulation and analysis of experimental results.

## 2. Principle and Modeling of the EMLA

### 2.1. The Operating Principle of the EMLA

In this study, a high-power density moving coil linear actuator developed by the research group is taken as the research object. The micro EMLA is composed of an inner yoke, an outer yoke, a permanent magnet array, a coil winding, and a coil bobbin. Among them, the permanent magnet array is arranged in the form of a Halbach array to enhance the air gap magnetic field strength. The adjacent coil windings are arranged in reverse order to reduce the influence of armature reaction effectively. It has the advantages of high-power density and rapid response, and its form is shown in Figure 1 [32].

The EMLA is a moving coil type voice coil motor. The novel EMLA has a simple structure consisting only of a moving coil winding and a magnetic yoke. Multiple layers of coils are wound on the coil skeleton to form the kinematics. It is more convenient to control the precise servo action of the kinematics indirectly by controlling the voltage at both ends of the winding [33].

### 2.2. Mathematical Modeling of EMLA

In summary, the EMLA is a complex system with the circuit subsystem, magnetic circuit subsystem, and mechanical subsystem coupled to one another, as shown in Figure 2. Its state equation is as follows [34]:(1)$$\left\{\begin{array}{l} \frac{dI(t)}{dt}=-\frac{R}{L}I(t)-\frac{K_{e}}{L}\;\dot{x}+\frac{U(t)}{L}\\ \;\ddot{x}=\frac{K_{m}}{M}I(t)-\frac{F_{f}\;+\;F_{d}}{M}\\ y=x \end{array}\right.$$
where *U*(*t*) is the desired voltage, *K*_e_ is the anti-electromotive force coefficient, *R* is resistance, *I*(*t*) is current, *L* is the inductance of the coil winding, *F*_e_ is the force, *K*_m_ is the electromagnetic force coefficient, *M* is the mass of the moving parts, *x* is the displacement, *F*_f_ is the frictional force, *F*_d_ is the disturbance, and *y* denotes the mover position.

Friction is one of the most prevalent nonlinear factors in high-precision servo systems. Particularly at low-velocity actuation, its complex characteristics have a particularly significant impact on high-precision positioning and tracking control [35]. Consequently, the actual operating conditions of the actuators are considered comprehensively in this study. The LuGre dynamic friction model is used to achieve accurate modeling of the actuator, which is defined as follows [36]:(2)$$\left\{\begin{array}{l} F_{f}=\sigma_{0}z+\sigma_{1}\frac{dz}{dt}+\sigma_{2}v\\ \frac{dz}{dt}=v-\frac{\left|v\right|}{g(v)}z\\ \sigma_{0}g(v)=F_{C}+(F_{S}-F_{C})e^{-{\left(v/v_{s}\right)}^{2}} \end{array}\right.$$
where $\sigma_{0},\sigma_{1}$, and $\sigma_{2}$ are the model parameters, *z* is an unmeasured internal friction state, *v* is the velocity, *v*_s_ is the experienced coefficient of the Stribeck, *F*_C_ is the Coulomb friction, *F*_S_ is the static friction.

From the above equation, the LuGre friction model has six unknown parameters, of which the static parameters are $\sigma_{2}$, *v*_s_, *F*_C_, and *F*_S_, and the dynamic parameters are $\sigma_{0}$ and $\sigma_{1}$. We conducted several actuator actuation experiments to investigate the static parameters and the dynamic parameters were estimated from the data of the pre-slip phase of the system, with detailed reference to the paper [37]. Finally, the offline parameter identification of the friction experimental data is performed based on a genetic algorithm (GA). The random sampling method is adopted as the selection operation method, the Gaussian operator is adopted for variational operation, and a uniform crossover operator is adopted for crossover operation. The finally obtained discrimination curve is shown in Figure 3. The fitted curve is a better match with the trend of the original experimental curve, the relative error is within 4.1%, and the identification results meet the desired requirements.

## 3. Design and Analysis of Control Method

### 3.1. Overall Control Method

The EMLA is the key factor in achieving high precision and high response for direct-drive servo systems. The goal is to achieve high response and high precision displacement control. Nevertheless, the EMLA is considered to drive the load directly. It is subjected to the direct action of various nonlinear and time-varying disturbances continuously. In particular, under complex operating conditions, strong disturbance and high-frequency noise disturbance of the captured signal make the performance significantly affected.

Consequently, an improved non-singular terminal sliding mode control (INTSMC) is designed based on an IERL. Concurrently, combined with the intelligent control advantages of fuzzy control to achieve adaptive adjustment of reaching law gain, to ensure that the response speed is improved while suppressing system jitter effectively. Finally, the effective observation compensation under noise disturbance is realized by the IDOB, and the whole control system framework is shown in Figure 4.

### 3.2. Design and Analysis of Improved Non-Singular Terminal Sliding Mode Control

Considering that the electrical time constant of the EMLA is much smaller than the mechanical time constant, the mathematical model Equation (1) is simplified to a second-order equation of state as shown in Equation (3):(3)$$\left\{\begin{array}{l} {\dot{x}}_{1}=x_{2}\\ {\dot{x}}_{2}=\frac{K_{m}}{M}u_{0}-\frac{F_{f}+F_{d}}{M}=\frac{K_{m}}{M}u_{0}+\frac{D}{M}\\ y=x_{1} \end{array}\right.$$
where *D* is the unknown nonlinear disturbance of the servo system, and the disturbance is assumed to be bounded in this study. $\left|D\right|\leq d$, *d* is a fixed constant.

Define the position versus velocity error of the EMLA as the following equation:(4)$$e_{y1}=y_{\mathrm{dir}}-y$$
(5)$$e_{y2}={\;\dot{e}}_{y1}={\;\dot{y}}_{\mathrm{dir}}-\;\dot{y}$$
where *y*_dir_ is the desired target and *y* is the actual displacement. *e*_y1_ and *e*_y2_ are the tracking error of actuator displacement and velocity, respectively.

In this study, to ensure fast reaching of the tracking error and simultaneously avoid the singularity of the conventional terminal sliding mode control, the non-singular terminal sliding mode control (NTSMC) is used, which is defined as [38]:(6)$$s(t)=e_{y1}+k_{1}{\left|e_{y1}\right|}^{\alpha }\mathrm{sign}(e_{y1})+k_{2}{\left|e_{y2}\right|}^{\beta }\mathrm{sign}(e_{y2})$$
where $k_{1},k_{2}\in R^{+}$, 1 < β < 2, 0 < α < β. We define β=p/q, *p*, and *q* are positive odd numbers. sign(·) is the sign function.

If the error reaches zero in finite time, then s(t)=0. We choose the Lyapunov function as $W=0.5{\left(e_{y1}\right)}^{2}$ the differentiation is given by the following equation:(7)$$\begin{array}{cc} \;\dot{W} & =e_{y1}e_{y2}\\ & =-e_{y1}k_{2}^{-\frac{1}{\beta }}\cdot \mathrm{sign}(e_{y1}){\left[e_{y1}+k_{1}{\left|e_{y1}\right|}^{\alpha }\mathrm{sign}(e_{y1})\right]}^{\frac{1}{\beta }}\\ & \leq -k_{2}^{\frac{1}{\beta }}\left|e_{y1}\right|(\left|e_{y1}\right|+k_{1}{\left|e_{y1}\right|}^{\alpha }{)}^{\frac{1}{\beta }}\leq 0 \end{array}$$

From the above analysis, we can conclude that the error *e*_y1_ will converge to zero in a fixed time. When away from equilibrium, a fast response is achieved based on $k_{1}{\left|e_{y1}\right|}^{\alpha }\mathrm{sign}(e_{y1})$. When approaching the equilibrium state, reaching in finite time is achieved based on $k_{2}{\left|e_{y2}\right|}^{\beta }\mathrm{sign}(e_{y2})$. Derivation of the above Equation (6) is obtained as:(8)$$\begin{array}{cc} \;\dot{s}(t) & =e_{y2}+{\alpha k}_{1}{\left|e_{y1}\right|}^{\alpha -1}e_{y2}+\beta k_{2}{\left|e_{y2}\right|}^{\beta -1}{\;\dot{e}}_{y2}\\ & =e_{y2}+{\alpha k}_{1}{\left|e_{y1}\right|}^{\alpha -1}e_{y2}+\beta k_{2}{\left|e_{y2}\right|}^{\beta -1}({\;\ddot{y}}_{\mathrm{dir}}-\;\ddot{y})\\ & =e_{y2}+\alpha k_{1}{\left|e_{y1}\right|}^{\alpha -1}e_{y2}+\beta k_{2}{\left|e_{y2}\right|}^{\beta -1}({\;\ddot{y}}_{\mathrm{dir}}-\frac{K_{m}}{M}u_{0}) \end{array}$$

The reaching law with excellent performance is also the key to the design of the sliding mode control. An IERL is obtained by introducing a power function of the error variable. It can greatly improve the convergence speed in the approach stage. Considering the high-frequency jitter caused by the discontinuous switching term in the traditional reaching law, the continuous function function(s) is used instead of the sign(s) to weaken the jitter. A comparison of the two functions is shown in Figure 5. By adjusting the relevant parameters of the continuous function(s), a continuous and smooth switching is achieved, which is of significance for the application of weakening jitter.

The IERL is defined as follows:(9)$$\left\{\begin{array}{l} \;\dot{s}=-\xi {\left|e_{y1}\right|}^{k_{3}}\mathrm{sign}(s)-\eta s\\ \underset{t\to \infty }{\lim}\left|e_{y1}\right|=0,\xi >0,\eta >0,k_{3}\geq 0 \end{array}\right.$$

For the sign(s) in the above equation, we replace it with the continuous function function(s) in this study. Further suppression of system high-frequency jitter can be achieved for a suitable value of *k*_4_. It is defined as follows:(10)$$\mathrm{function}(s)=\frac{{\left(1+s\right)}^{k_{4}}-{\left(1-s\right)}^{k_{4}}}{{\left(1+s\right)}^{k_{4}}+{\left(1-s\right)}^{k_{4}}}$$

By combining the exponential reaching laws of (8) and (9), the u0 is obtained as:(11)$$u_{0}=u_{\mathrm{eq}}+u_{\mathrm{asw}}$$
(12)$$u_{\mathrm{eq}}=\frac{M}{K_{m}}\left[{\;\ddot{y}}_{\mathrm{dir}}+\frac{{\left|e_{y2}\right|}^{2-\beta }}{\beta k_{2}}\left(1+{\alpha k}_{1}{\left|e_{y1}\right|}^{\alpha -1}\right)\mathrm{sign}(e_{y2})\right]$$
(13)$$u_{\mathrm{asw}}=\frac{M}{K_{m}}\left(\xi {\left|e_{y1}\right|}^{k_{3}}\mathrm{function}(s)+\eta s\right)$$

For the improved controller stability analysis, the Lyapunov function is chosen as:(14)$$V=\frac{1}{2}s^{2}$$

Taking the derivative of the above Equation (14), we can obtain:(15)$$\begin{array}{ll} \dot{V} & =s\dot{s}=s\cdot \left[e_{y2}+\alpha k_{1}{\left|e_{y1}\right|}^{\alpha -1}e_{y2}+\beta k_{2}{\left|e_{y2}\right|}^{\beta -1}({\ddot{y}}_{\mathrm{dir}}-\frac{K_{m}}{M}u_{0}-\frac{D}{M})\right]\\ & =s\cdot \left[e_{y2}+\alpha k_{1}{\left|e_{y1}\right|}^{\alpha -1}e_{y2}+\beta k_{2}{\left|e_{y2}\right|}^{\beta -1}({\ddot{y}}_{\mathrm{dir}}-\frac{K_{m}}{M}(u_{\mathrm{eq}}+u_{\mathrm{asw}})-\frac{D}{M})\right]\\ & =s\cdot \left[e_{y2}+\alpha k_{1}{\left|e_{y1}\right|}^{\alpha -1}e_{y2}-\left|e_{y2}\right|\mathrm{sign}(e_{y2})\left(1+{\alpha k}_{1}{\left|e_{y1}\right|}^{\alpha -1}\right)-\beta k_{2}{\left|e_{y2}\right|}^{\beta -1}(\frac{K_{m}}{M}u_{\mathrm{asw}}-\frac{D}{M})\right]\\ & =-s\cdot \left[\beta k_{2}{\left|e_{y2}\right|}^{\beta -1}(\frac{K_{m}}{M}u_{\mathrm{asw}}-\frac{D}{M})\right]\\ & =-\beta k_{2}{\left|e_{y2}\right|}^{\beta -1}\cdot \left[\xi {\left|e_{y1}\right|}^{k_{3}}\mathrm{function}(s)s+\eta s^{2}-\frac{D}{M}s\right]\\ & \leq -\beta k_{2}{\left|e_{y2}\right|}^{\beta -1}\cdot \left[\xi {\left|e_{y1}\right|}^{k_{3}}\left|s\right|+\eta s^{2}-\frac{d}{M}\left|s\right|\right]\\ & \leq -\beta k_{2}{\left|e_{y2}\right|}^{\beta -1}\cdot \left[\left|s\right|{\left|e_{y1}\right|}^{k_{3}}(\xi -\frac{d}{{\left|e_{y1}\right|}^{k_{3}}M})+\eta s^{2}\right]\leq 0 \end{array}$$

With the above equation, we know that there exists ξ, such that holds $\xi >d\cdot {\left({\left|e_{y1}\right|}^{k_{3}}M\right)}^{-1}$. At this moment, it is known that the proposed controller is stable and the upper bound of the system reaching time depends on the gain parameter of the IERL. The parameters of the controller can be reasonably selected based on the trade-off between convergence speed and steady-state tracking accuracy [39].

### 3.3. Adaptive Non-Singular Terminal Sliding Mode Control

To further improve the adaptability and response performance of the EMLA under complex operating conditions, the robustness and fast response performance of the balanced control system can be achieved by considering the gain parameters ξ and η [40]. Concurrently, to ensure the rapid reaching of the system state to the sliding mode surface, the adaptive non-singular terminal sliding mode control (ANTSMC) is designed by combining the advantages of sliding mode control and fuzzy control under the premise of ensuring the stability of the controller, which realizes the self-adjustment and self-adaptation of the controller parameters. The fuzzy reaching law is designed as follows:(16)$$\;\dot{s}=-\xi {\left|e_{y1}\right|}^{k_{3}}\mathrm{sat}(s)-\eta s=-\xi_{0}\xi_{i}{\left|e_{y1}\right|}^{k_{3}}\mathrm{sat}(s)-\eta_{0}\eta_{i}s$$
where $\xi_{0}$ and $\eta_{0}$ are the initial values. $\xi_{i}$ and $\eta_{i}$ are the fuzzy control outputs.

Based on the above considerations, $s\dot{s}$ is designed as input, $\xi_{i}$ and $\eta_{i}$ as output, and fuzzy control rules between input and output are developed as shown in Table 1. The membership functions of the input and output are shown in Figure 6 and Figure 7. In this study, Mamdani fuzzy inference rules are adopted and based on the center of gravity method centroid to solve the fuzzy, to realize the switching term in the control law into a continuous fuzzy output. It is guaranteed to achieve a good control performance even under time-varying parameters and different working conditions of the EMLA.

### 3.4. Design and Analysis of Disturbance Observer

To meet the control requirements of the EMLA for high-quality response under severe conditions. The disturbance observer is introduced to observe and compensate the servo system, and the ANTSMC and the disturbance observation and compensation technique are combined to further improve the anti-disturbance capability. The observation and compensation of large external nonlinear disturbances and unmodeled factors of the servo system are realized.

Simultaneously, considering that the designed disturbance observer needs to estimate the displacement and velocity signals, which are affected by the noise of the acquired signal and the displacement sampling frequency. The velocity signal obtained by directly differentiating the displacement signal with noise is amplified and quantized, which affects the robustness of the controller seriously. Consequently, the TD is introduced to obtain derivatives with good displacement signals and avoid amplification of noise [41]. The tracking error is expressed as:(17)$$\left\{\begin{array}{l} \;e_{y1}=y_{\mathrm{dir}}-y\\ {\;\dot{e}}_{\mathrm{tdy}1}={\;\dot{y}}_{\mathrm{dir}}-{\;\dot{\hat{y}}}_{\mathrm{td}} \end{array}\right.$$
where ${\;\dot{\hat{y}}}_{\mathrm{td}}$ is the estimated velocity of the TD; the TD is expressed as:(18)$$\left\{\begin{array}{l} {\;\dot{\hat{y}}}_{\mathrm{td}}=\varphi \\ \;\dot{\varphi }=\mathrm{fhan}({\;\hat{y}}_{\mathrm{td}}(\tau )-y(\tau ),{\;\dot{\hat{y}}}_{\mathrm{td}}(\tau ),r,h_{0}) \end{array}\right.$$
where *r* and *h*_0_ are parameters of the tracking differentiator. $\mathrm{fsg}(\rho ,d_{0})$ is expressed as:(19)$$\mathrm{fsg}(\rho ,d_{0})=\frac{\mathrm{sign}(\rho +d_{0})-\mathrm{sign}(\rho -d_{0})}{2}$$
where fhan function is defined as follows:(20)$$\left\{\begin{array}{l} d_{0}={rh}_{0}{}^{2}\\ a_{0}=h_{0}{\;\dot{\hat{y}}}_{\mathrm{td}}(\tau )\\ y_{0}={\;\hat{y}}_{\mathrm{td}}(\tau )-y(\tau )+a_{0}\\ a_{1}=\sqrt{d_{0}{}^{2}+8d_{0}\left|y_{0}\right|}\;\\ a_{2}=a_{0}+\mathrm{sign}(y_{0})\cdot (a_{1}-d_{0})/2\\ a=(a_{0}+y)\cdot \mathrm{fsg}(y_{0},d_{0})+a_{2}(1-\mathrm{fsg}(y_{0},d_{0}))\\ \mathrm{fhan}=-r(\frac{a}{d_{0}})\mathrm{fsg}(a,d_{0})-r\mathrm{sign}(a)(1-\mathrm{fsg}(a,d_{0})) \end{array}\right.$$

Viewing the perturbation in Equation (3) as a state variable, the model can be described in the form of an extended state equation.
(21)$$\left\{\begin{array}{l} {\;\dot{x}}_{1}=x_{2}\\ {\;\dot{x}}_{2}=\frac{K_{m}}{M}u+\frac{D}{M}\\ \;\dot{D}=d(t) \end{array}\right.$$

Because the disturbance is bounded, we get $\left|\;\dot{D}\right|=\left|d(t)\right|\leq d_{2}$. The designed disturbance observer is obtained as follows:(22)$$\left\{\begin{array}{l} \;e_{\mathrm{oy}1}={\;\hat{x}}_{2}-{\;\dot{\hat{y}}}_{\mathrm{td}}\\ {\;\dot{\hat{x}}}_{2}=\frac{K_{m}}{M}u+\frac{\;\tilde{D}}{M}-\lambda_{1}{\left|e_{\mathrm{oy}1}\right|}^{\frac{1}{2}}\mathrm{sign}(e_{\mathrm{oy}1})\\ \;\dot{\tilde{D}}=-\lambda_{2}\mathrm{sign}(e_{\mathrm{oy}1}) \end{array}\right.$$
where $\;\tilde{D}$ is the observed value of the disturbance, $\lambda_{1}$ and $\lambda_{2}$ are the positive gain control parameters of the observer. The error equation can be obtained by subtracting Equation (21) from Equation (22) as:(23)$$\left\{\begin{array}{l} \;e_{\mathrm{oy}3}=\;\tilde{D}-D\\ {\;\dot{e}}_{\mathrm{oy}2}=\frac{e_{\mathrm{oy}3}}{M}-\lambda_{1}{\left|e_{\mathrm{oy}2}\right|}^{\frac{1}{2}}\mathrm{sign}(e_{\mathrm{oy}2})\\ {\;\dot{e}}_{\mathrm{oy}3}=-\lambda_{2}\mathrm{sign}(e_{\mathrm{oy}2})-d(t) \end{array}\right.$$

Define the following vectors:(24)$$\eta ={\left[\eta_{1},\eta_{2}\right]}^{T}={\left[{\left|e_{\mathrm{oy}2}\right|}^{\frac{1}{2}}\mathrm{sign}(e_{\mathrm{oy}2}),e_{\mathrm{oy}3}\right]}^{T}$$

Then, we can obtain:(25)$${\left\|\eta \right\|}_{2}=\sqrt{\eta_{1}{}^{2}+\eta_{2}{}^{2}}\geq \left|\eta_{1}\right|$$

Define Lyapunov function as $V_{1}=\eta^{T}P\eta$, the P is defined as:(26)$$P=\frac{1}{2}\left[\begin{array}{cc} \lambda_{1}{}^{2}+4\lambda_{2} & -\lambda_{1}\\ -\lambda_{1} & 2 \end{array}\right]$$

Then, we can obtain:(27)$$\begin{array}{c} {\;\dot{V}}_{1}=-\frac{1}{2\left|\eta_{1}\right|}\eta^{T}Q\eta +\zeta^{T}\eta d(t)\\ \;\;\;\;\;\;\;\;\;\;\;\;\;\;\;\;\;\;\leq -\left(\frac{1}{2}\varsigma_{\min}\left\{Q\right\}-d_{2}{\left\|\zeta \right\|}_{2}\right){\left\|\eta \right\|}_{2}\leq 0 \end{array}$$
where $\varsigma_{\min}\left\{Q\right\}$ is the minimum eigenvalue of Q, $\zeta ={\left[-\lambda_{1},2\right]}^{T}$, the symmetric positive definite matrix Q is defined as:(28)$$Q=\left[\begin{array}{cc} \lambda_{1}{}^{3}+2\lambda_{1}\lambda_{2} & -\lambda_{1}{}^{2}\\ -\lambda_{1}{}^{2} & \lambda_{1} \end{array}\right]$$

When the parameters of the controller are properly selected, the observer will accurately track the disturbance in a finite time, i.e., the observation error will converge to the origin in a finite time [42].

## 4. Experimental Verification and Analysis

### 4.1. Experimental Equipment

To further verify the effectiveness of the proposed control, the EMLA performance experimental platform is built for test verification, and the test platform as shown in Figure 8 is built. The hardware circuit diagram is shown in Figure 9. An integrated controller based on the 32-bit floating-point digital signal processor with a dominant frequency of 300 MHz is used to realize the control algorithm and acquisition signal processing operation. The displacement is obtained by the displacement sensor PM11-3, and the accuracy of the displacement sensor is 0.001 mm. The current is collected by the current sensor ACS712, and its accuracy is 0.05%. The stroke of the EMLA is 10 mm, the resistance is 1.5 Ω, and the dynamic mass is 0.256 kg.

### 4.2. Analysis of Step Target Results

To verify the performance of the ANTSMC in this study in realizing the fast response and accurate actuation of the EMLA. The simulation and experimental results of the proposed control are compared with the conventional non-singular terminal sliding mode control (NTSMC) and PID. The response performance of different controls is evaluated by the integral of time and absolute error (ITAE) criteria and the integral absolute error (IAE) criteria. The ITAE criteria and the IAE criteria are expressed as follows.
(29)$$J_{\mathrm{ITAE}}=\int_{0}^{\infty }t\left|e(t)\right|dt$$
(30)$$J_{\mathrm{IAE}}=\int_{0}^{\infty }\left|e(t)\right|dt$$

Under the 5 mm step target, the response results of different controls are shown in Figure 10 and Figure 11 respectively. The simulation results showed that different controls responded quickly under the step expected signal without overshoot tracking. Compared with PID and the NTSMC, the ANTSMC had higher steady-state accuracy, faster response speed, and better static response performance. The experimental results showed that both the NTSMC and PID had a certain overshoot, whereas the ANTSMC had almost no overshoot and a fast response. The experimental response time of the ANTSMC also approximated the simulation results, and the effectiveness of the advanced and refined modeling of the control in the simulation was verified.

The experimental results of different controls showed that the PID had the largest overshoot with a maximum deviation of 0.185 mm and an overshoot of 2.8%. The maximum deviation of the NTSMC was 0.104 mm and the overshoot was 2.74%, whereas ANTSMC could achieve effective observation and compensation for nonlinear factors such as disturbances in the experiments, which could be seen to be nearly overshoot-free and responded quickly. At the same time, the experimental response curve of the ANTSMC was also in good agreement with the simulation results, which verified the effectiveness of the accurate modeling and provided an effective method for the accurate tracking control of the direct-drive servo system.

The response times of the different controls under simulation and experiment are shown in Table 2. The response times of PID and the NTSMC under simulation were 11.12 ms and 10.9 ms, respectively, with steady-state errors less than 3 μm. The response time under the simulation of the ANTSMC was only 9.63 ms, which was reduced by 15.47% and 13.19% compared with the other two controls, respectively, and the steady-state error of the ANTSMC was 0 mm. This was because the non-singular terminal sliding mode surface (NTSMS) and the IERL further improved the system control accuracy while ensuring the fast reaching of the servo system tracking error in a finite time.

In the experiments, the response times of PID and the NTSMC were 12.25 ms and 11.04 ms, respectively. The steady-state errors were −0.046 mm and 0.032 mm, respectively, but both controls had overshoot. The response time of ANTSMC was only 9.85 ms, which was 24.36% and 12.08% less than PID and the NTSMC respectively, and the steady-state error was only 5 μm. Considering the limitation of the actual DC supply voltage and the limitation of the introduced differential tracker by the solution step, the response time and the steady-state accuracy of the controls were somewhat different compared with the simulation results. Nevertheless, it was able to meet the control requirements of the EMLA in terms of response speed and steady-state accuracy of the drive unit under the actual complex working conditions.

The error, input voltage signal, and current curves of the ANTSMC are shown in Figure 12. In the initial moment of the EMLA, the error of displacement was large, and under the effect of feedback from the control, the voltage became large rapidly and then started to decrease to 0 with the rapid approach of the actuator to the desired target. Concurrently, the current in the coil winding began to increase, but at a slower rate than the voltage rise. This was because the effect of the counter-electromotive force prevented the flow of current in the coil, and the current began to decrease after reaching its maximum value and eventually converged to a steady-state.

In summary, the comparative analysis of the response curves in Figure 10, Figure 11 and Figure 12 showed that the ANTSMC was generally better than the PID and NTSMC in terms of response speed, control accuracy, and stability in the experiment under the step expectation target. ANTSMC had a better dynamic response and steady-state effect, and the effectiveness of the ANTSMC was effectively verified.

### 4.3. Analysis of Non-Repetitive Target Displacement Following

Considering that the EMLA needs to accurately track targets with different amplitudes under different working conditions, to verify the tracking response performance of the proposed control in a non-repetitive target trajectory. Simultaneously, considering that under complex working conditions, the servo system is often disturbed by the signal noise collected by the sensor. To further test the tracking response performance of the proposed control for the non-repetitive target trajectory under the disturbance of continuous acquisition noise, the non-repetitive expected target was set, and a certain power of white noise was superimposed at the feedback of the displacement sensor. In this section, an NTSMC based on a disturbance observer (DOB) was added for comparison. The non-repetitive target response curves of different controls are shown in Figure 13.

It can be seen from the tracking trajectory response diagram in Figure 13 that different controls could better track the desired target. Due to the disturbance of noise, the NTSMC based on the DOB, NTSMC, and PID produced certain irregular fluctuations in the response and further led to the phase lag of their response. Among them, it could be seen that the response of PID had the largest lag. The ANTSMC has strong robustness due to the tracking differentiator. At the same time, because of its combination with the advantages of fuzzy control, it could track the non-repetitive target quickly and effectively, as well as avoid the noise disturbance introduced by the traditional differentiator directly. It ensured the effective observation of the disturbance observer and further ensured the response quality of the servo system.

The displacement profile of the ANTSMC with white noise disturbance, the actual response, and the velocity estimation of the tracking differentiator are shown in Figure 14. The ANTSMC achieved a better estimation of velocity by tracking differentiators under the disturbance of white noise and ensuring the effective operation of the DOB. Although there was a loss in the initial response, the response could still be better achieved.

The comparison curves of displacement errors for different controls are shown in Figure 15. Compared to the response performance of the NTSMC based on the DOB, NTSMC, and PID, ANTSMC could quickly converge to the target because of the IERL introduced. Simultaneously, the tracking response performance of ANTSMC was significantly better than that of PID, NTSMC, and the NTSMC based on the DOB throughout the tracking process because the introduced fuzzy control realized the adaption of the key parameters of the IERL. In particular, the maximum tracking errors of PID, NTSMC, and the NTSMC based on the DOB were 0.31 mm, 0.081 mm, and 0.078 mm, respectively, while the maximum tracking error of ANTSMC was less than 0.01 mm.

As shown in Figure 16 for the evaluation results under different controls, it could be seen that the evaluation results of ANTSMC under the ITAE criterion as well as the IAE criterion were much smaller than the other controls during the whole non-repetitive desired target tracking process. It was only 1.6 × 10^−4^ and 8.92 × 10^−3^, respectively. Under the ITAE evaluation criteria, ANTSMC improved to 4.17% for PID and 22.54% for NTSMC. The evaluation results of the NTSMC based on the DOB were larger than the evaluation values of ANTSMC as well as NTSMC. This was because the noise was further amplified through the differential of the displacement, which led to the failure of the DOB compensation and the system response quality was severely affected. In summary, the comparative analysis showed that ANTSMC could not only improve the response speed of the system, but could also track the non-repetitive desired target displacement with higher accuracy. At the same time, ANTSMC could achieve fast and smooth tracking of the desired target under continuous noise disturbance, which also further verified the effectiveness of ANTSMC proposed in this study.

### 4.4. Robust Performance Analysis

To verify the effectiveness of the proposed ANTSMC based on disturbance observer, the suppression ability of the servo system for fixed load disturbance is further studied. The expected target of the 9 mm step is set and tested under fixed load conditions of 0 N, 10 N, 20 N, and 30 N. The results of PID, NTSMC, and ANTSMC proposed in this study are compared and analyzed. The response curves under different load conditions are shown in Figure 17. It can be seen from Figure 17 that under the load conditions of 0 N, 10 N, 20 N, and 30 N, the ANTSMC quickly tracked the response under the given step signal, and the tracking curves of different loads are in good agreement. Furthermore, due to the existence of the DOB, fast observation compensation for different loads was realized. In the whole response process, the *J*_ITAE_ value was the smallest, and the observation steady-state error of the disturbance observer to the load was kept within 1.02%. The effectiveness of the designed observer was verified.

The robustness of controls under different load conditions was evaluated according to ITAE criteria. The response performance evaluation results of different controls are shown in Figure 18. Among them, the evaluation results of the ANTSMC under the ITAE criterion were lower than the other controls. Concurrently, with the continuous increase of load, the *J*_ITAE_ values of different controls increased, and it was obvious that the JITAE values under PID changed rapidly. However, the *J*_ITAE_ value under the ANTSMC increased slowly compared with other controls, which fully showed that the control system had strong robustness. Concurrently, under no-load conditions, the *J*_ITAE_ value under PID control was slightly smaller than that under NTSMC. This was because the discontinuous switching of sign function in the NTSMC reduced the steady-state accuracy and response quality of the servo system.

In conclusion, the *J*_ITAE_ value of the ANTSMC was unchanged with the increase of load and was smaller than the other controls. This clearly showed that the robustness of the control system based on the ANTSMC proposed in this study was better than PID and NTSMC. It also verified the effectiveness and progressiveness of the ANTSMC proposed in this study.

## 5. Conclusions

In this study, an ANTSMC based on an IDOB is proposed to improve the response performance of an electromagnetic linear actuator for different targets under severe operating conditions.

(1) The INTSMC is designed based on the IERL. The tracking error convergence in finite time is ensured while the tracking error convergence speed is improved. The stability of the system is analyzed based on Lyapunov function. The analysis and results show that the closed-loop control system is stable and the errors converge in finite time.

(2) Fuzzy control is combined with the advantage of INTSMC to solve the problems, such as insufficient adaptive adjustment ability of convergence law gain. Under the step response, the proposed control method was reduced by 24.36% and 12.08% respectively compared with other controls, and the steady-state error was only 5 μm.

(3) An IDOB is designed based on the tracking differentiator to ensure effective observation under complex operating conditions. Simulation and experimental results show that the ANTSMC still has strong robustness and excellent response performance under highly complex operating conditions.

## Figures and Tables

**Figure 1 micromachines-13-01294-f001:**
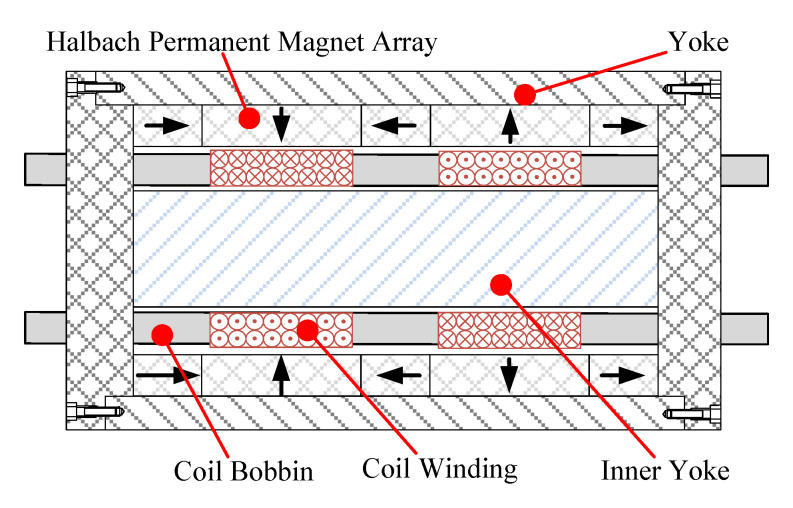
Structure schematic of EMLA.

**Figure 2 micromachines-13-01294-f002:**
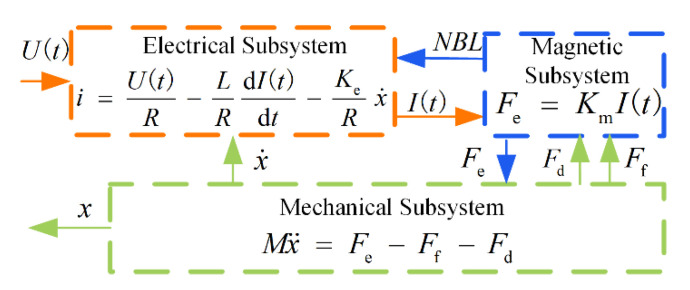
The Coupling diagram of EMLA.

**Figure 3 micromachines-13-01294-f003:**
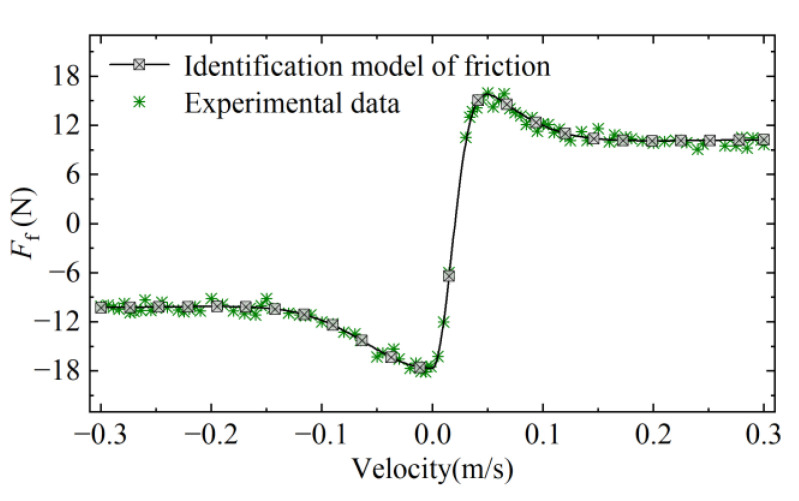
Relationship between friction and velocity.

**Figure 4 micromachines-13-01294-f004:**
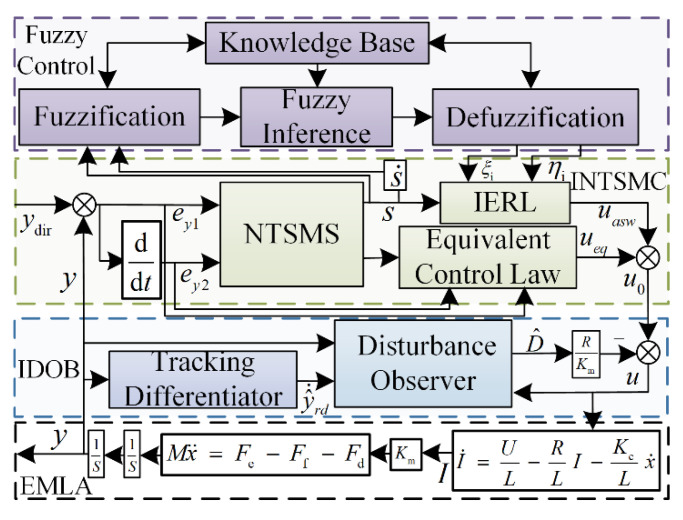
The framework of the control system.

**Figure 5 micromachines-13-01294-f005:**
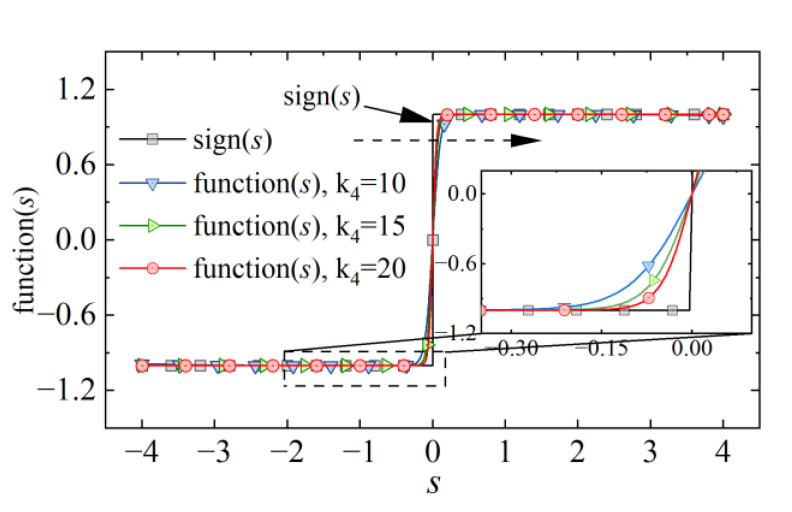
Comparison curves of function(s) and sign.

**Figure 6 micromachines-13-01294-f006:**
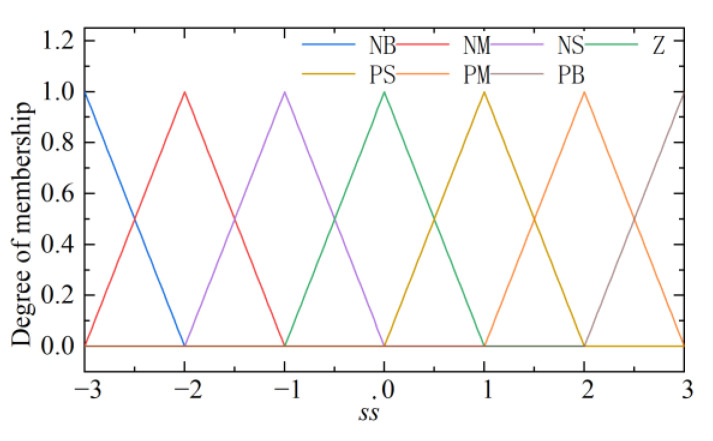
Input $s\dot{s}$.

**Figure 7 micromachines-13-01294-f007:**
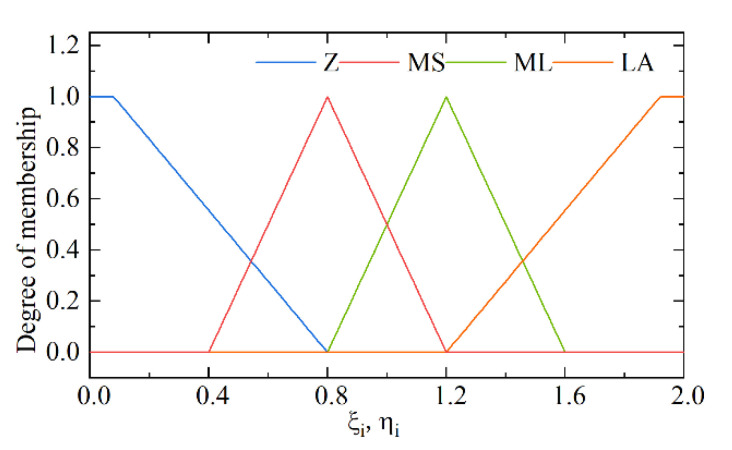
Outputs $\xi_{i},\eta_{i}$.

**Figure 8 micromachines-13-01294-f008:**
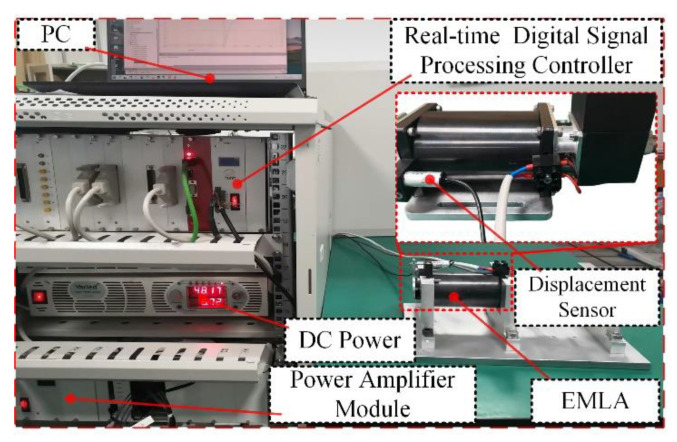
The experimental platform of the EMLA.

**Figure 9 micromachines-13-01294-f009:**
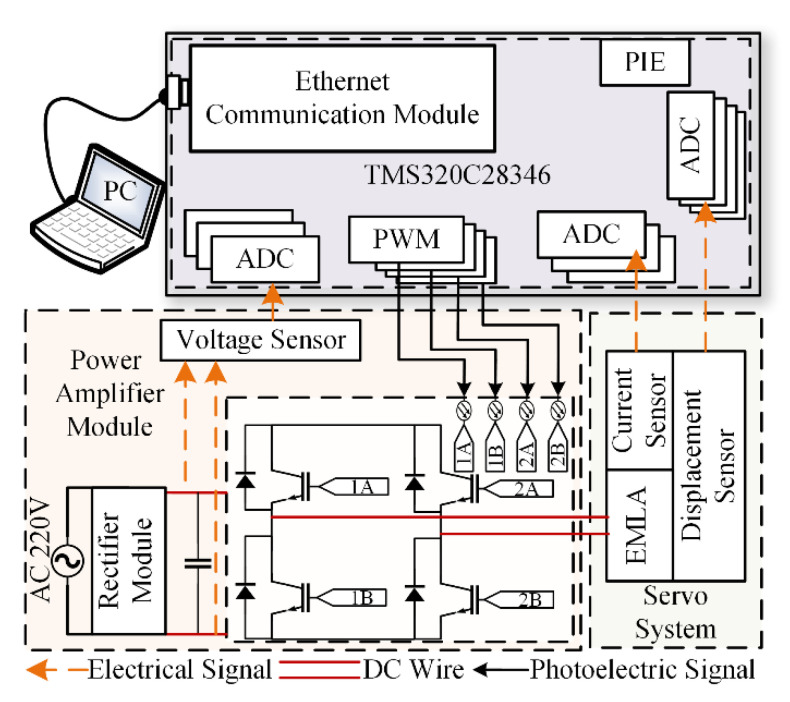
Hardware circuit diagram of the platform.

**Figure 10 micromachines-13-01294-f010:**
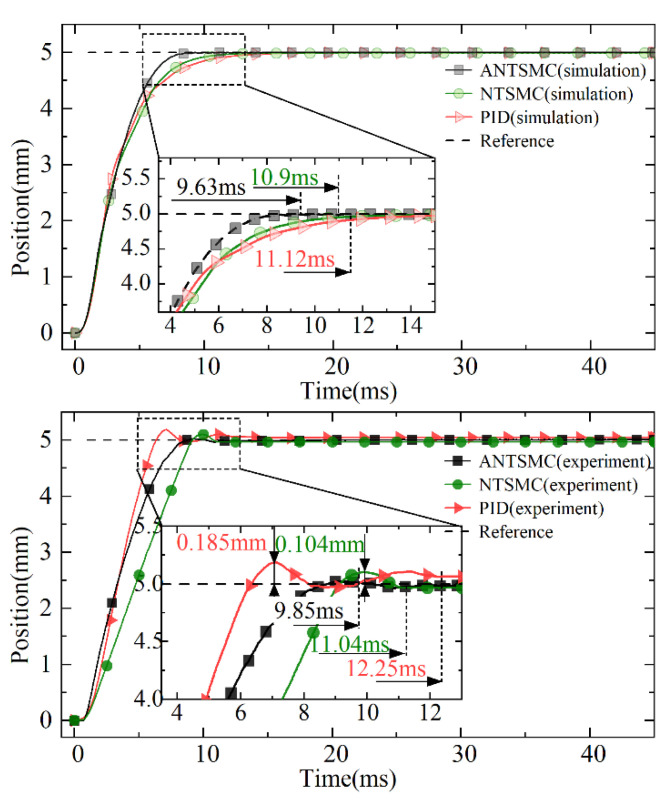
Response curves of displacement under different controls.

**Figure 11 micromachines-13-01294-f011:**
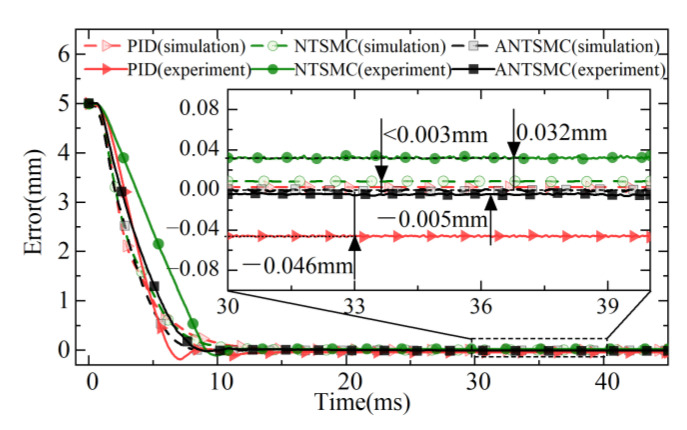
Errors in tracking.

**Figure 12 micromachines-13-01294-f012:**
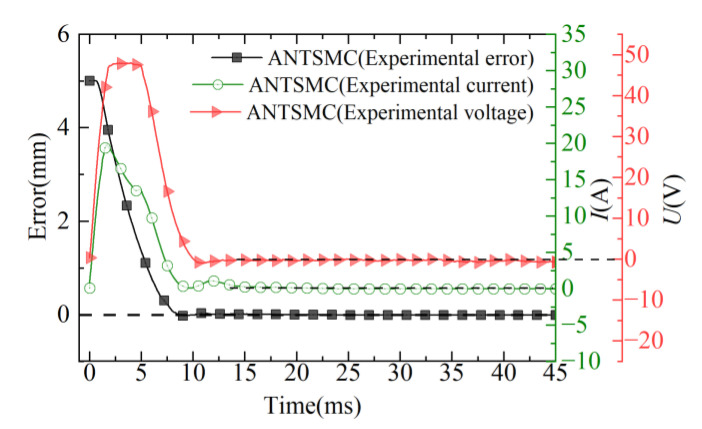
Curves of error, current, and voltage.

**Figure 13 micromachines-13-01294-f013:**
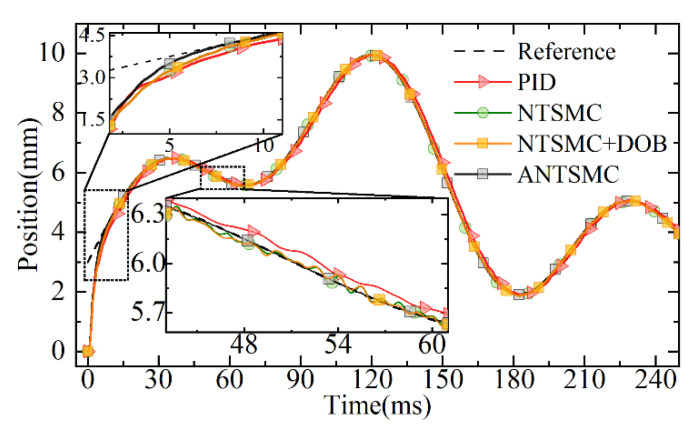
The response curves of the non-repetitive target.

**Figure 14 micromachines-13-01294-f014:**
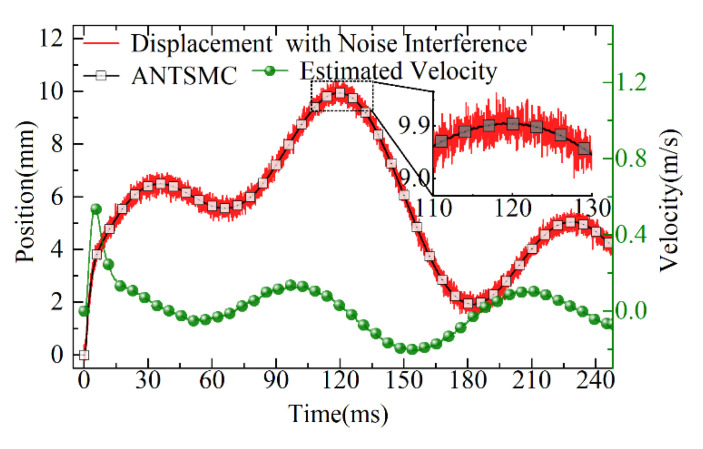
The curves of tracking and velocity estimation.

**Figure 15 micromachines-13-01294-f015:**
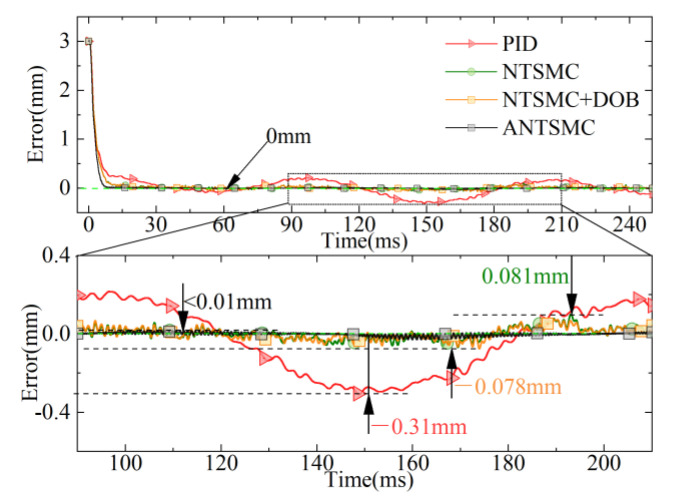
Tracking error of different controls.

**Figure 16 micromachines-13-01294-f016:**
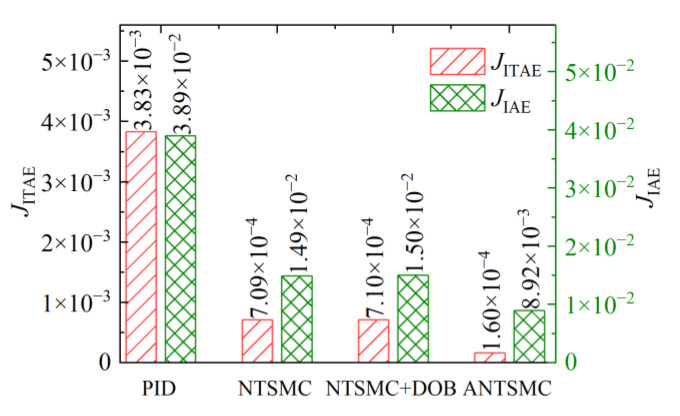
Response evaluation of different controls.

**Figure 17 micromachines-13-01294-f017:**
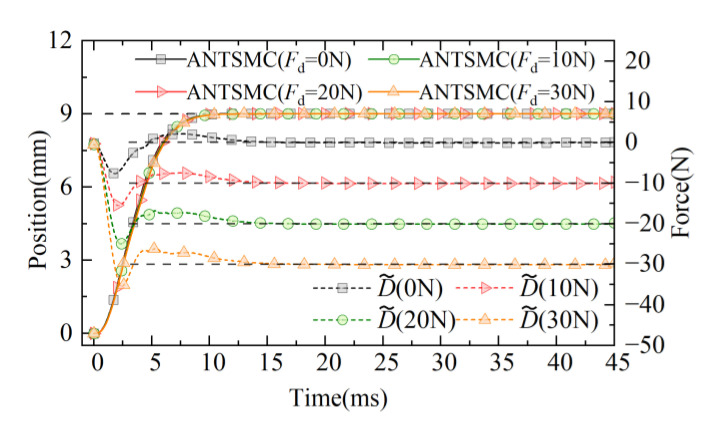
Response curves of displacement and disturbance observation.

**Figure 18 micromachines-13-01294-f018:**
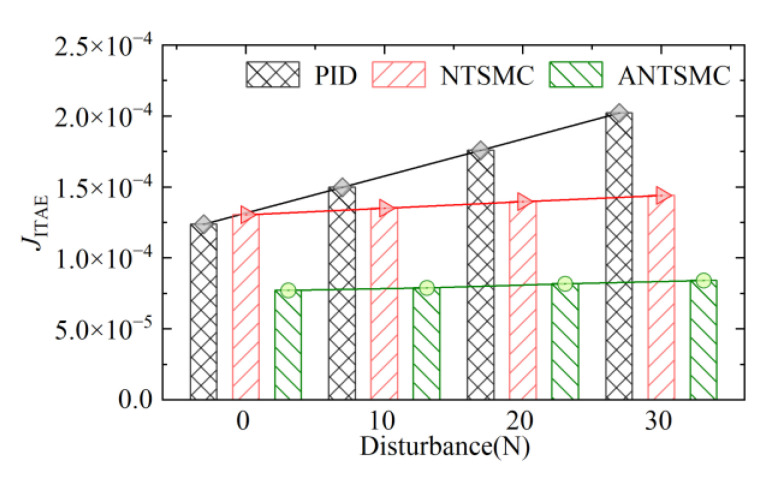
Performance evaluation of different controls.

**Table 1 micromachines-13-01294-t001:** Fuzzy control rules.

Input	NB	NM	NS	ZO	PS	PM	PB
*ξ* _i_	LA	ML	MS	Z	MS	ML	LA
*η* _i_	LA	ML	MS	Z	MS	ML	LA

**Table 2 micromachines-13-01294-t002:** Response time of the step target.

Controls	Simulation Results	Experimental Results
PID	11.12 ms	12.25 ms
NTSMC	10.9 ms	11.04 ms
ANTSMC	9.63 ms	9.85 ms

## Data Availability

The data presented in this study are available on request from the corresponding author.

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
