# Peer review of "Adaptive Non-Singular Terminal Sliding Mode Control Method for Electromagnetic Linear Actuator"

_micromachines, 2022, doi:10.3390/mi13081294_

Round 1
Reviewer 1 Report
The manuscript is properly organized and seems novel to me.
However, I have some remarks and suggestions:
1. Considering system (1), the authors introduced variable $y$, but there is no description of the $y$-variable in the following text;
2. The transition from equation (1) to equation (3), in my opinion, must be done within the small parameter approach. System (3) can be obtained from (1) at $L\arrow\inf$, however the correctness of the transition must be approved;
3. In page 5, equations (4) and (5) are the same;
4. First time in page 6 and in the following text there are many times the authors use term ``Lyapunov’’ instead of the ``Lyapunov function’’. Of course, this must be improved;
5. In the main text there is no relation between $e_{1y}$ and $e_{2y}$, thus equation (7) seems strange;
6. In page (9) in equation (18) the authors must check the notation of the right-hand side;
7. There many phrases that must be paraphrased because of troubles for understanding
In my opinion, the manuscript seems fresh and can be published after minor revision.
Author Response
Dear reviewer:
Thank you for your decision and constructive comments on my manuscript. We have carefully considered the suggestion of the Reviewer and made some changes. We have tried our best to improve and made some changes in the manuscript.
The yellow part has been revised according to your comments.
Please see the attachment.

Reviewer 2 Report
This paper studied an adaptive non-singular terminal sliding mode controller to improve the response performance of IERL, which presented sufficient work on both theoretical and experimental efforts.
Authors should pay attention to the following points:
1. The figure 2 depicts the connections coupled of the subsystems, it’s recommended that authors should indicate the term in the mechanical subsystem directly for clarity.
2. The figure 4 is suggested to enlarge to a proper scale including the symbols and connections within for a clearer demonstration.
3. It’s confused that if there is a symbol mistake in equation 18. Please confirm the correction.
4. The simulation results depicted in figure 10 and 15 look messy. It’s suggested to make it clearer in another way for comparisons.
5. As it’s mentioned in the introduction, author assumed that the disturbances is caused by a nonlinear noise. However, as it’s shown in figure 17 and 18, the disturbance is set as a fixed force in the analysis. The author should explain clearly about this mismatch or give relevant support.
6. The main background of this paper is not clear. The authors should highlight the main background of this paper. The background information is very useful for the understanding of this paper. Some new advances in this field are missing, such as 10.1109/TAES.2021.3103258.
7. The conclusion 1 is incomplete.
Above all, the paper can be accepted after minor revisions.
Author Response

(The authors gave the same response as above.)
